# Effect of Important Food Sources of Fructose-Containing Sugars on Inflammatory Biomarkers: A Systematic Review and Meta-Analysis of Controlled Feeding Trials

**DOI:** 10.3390/nu14193986

**Published:** 2022-09-26

**Authors:** XinYe Qi, Laura Chiavaroli, Danielle Lee, Sabrina Ayoub-Charette, Tauseef A. Khan, Fei Au-Yeung, Amna Ahmed, Annette Cheung, Qi Liu, Sonia Blanco Mejia, Vivian L. Choo, Russell J. de Souza, Thomas M. S. Wolever, Lawrence A. Leiter, Cyril W. C. Kendall, David J. A. Jenkins, John L. Sievenpiper

**Affiliations:** 1Department of Nutritional Sciences, Temerty Faculty of Medicine, University of Toronto, Toronto, ON M5S1A8, Canada; 2Toronto 3D Knowledge Synthesis and Clinical Trials Unit, Clinical Nutrition and Risk Factor Modification Centre, St. Michael’s Hospital, Toronto, ON M5C 2T2, Canada; 3Department of Family and Community Medicine, University of Toronto, Toronto, ON M5G1V7, Canada; 4Department of Health Research Methods, Evidence, and Impact, Faculty of Health Sciences, McMaster University, Hamilton, ON L8S4K1, Canada; 5Population Health Research Institute, Hamilton Health Sciences Corporation, Hamilton, ON L8L2X2, Canada; 6INQUIS Clinical Research Ltd. (Formerly GI Labs), Toronto, ON M5C2N8, Canada; 7Department of Medicine, Temerty Faculty of Medicine, University of Toronto, Toronto, ON M5S1A8, Canada; 8Division of Endocrinology and Metabolism, Department of Medicine, St. Michael’s Hospital, Toronto, ON M5C 2T2, Canada; 9Li Ka Shing Knowledge Institute, St. Michael’s Hospital, Toronto, ON M5B1T8, Canada; 10College of Pharmacy and Nutrition, University of Saskatchewan, Saskatoon, SK S7N5E5, Canada

**Keywords:** inflammation, CRP, fructose, sugars, food sources, fruit, fruit juice, sugar-sweetened beverages, systematic review, meta-analysis

## Abstract

Background: Fructose-containing sugars as sugar-sweetened beverages (SSBs) may increase inflammatory biomarkers. Whether this effect is mediated by the food matrix at different levels of energy is unknown. To investigate the role of food source and energy, we conducted a systematic review and meta-analysis of controlled trials on the effect of different food sources of fructose-containing sugars on inflammatory markers at different levels of energy control. Methods: MEDLINE, Embase, and the Cochrane Library were searched through March 2022 for controlled feeding trials ≥ 7 days. Four trial designs were prespecified by energy control: substitution (energy matched replacement of sugars); addition (excess energy from sugars added to diets); subtraction (energy from sugars subtracted from diets); and ad libitum (energy from sugars freely replaced). The primary outcome was *C*-reactive protein (CRP). Secondary outcomes were tumour necrosis factor-alpha (TNF-α) and interleukin-6 (IL-6). Independent reviewers extracted data and assessed risk of bias. GRADE assessed certainty of evidence. Results: We identified 64 controlled trials (91 trial comparisons, *n* = 4094) assessing 12 food sources (SSB; sweetened dairy; sweetened dairy alternative [soy]; 100% fruit juice; fruit; dried fruit; mixed fruit forms; sweetened cereal grains and bars; sweets and desserts; added nutritive [caloric] sweetener; mixed sources [with SSBs]; and mixed sources [without SSBs]) at 4 levels of energy control over a median 6-weeks in predominantly healthy mixed weight or overweight/obese adults. Total fructose-containing sugars decreased CRP in addition trials and had no effect in substitution, subtraction or ad libitum trials. No effect was observed on other outcomes at any level of energy control. There was evidence of interaction/influence by food source: substitution trials (sweetened dairy alternative (soy) and 100% fruit juice decreased, and mixed sources (with SSBs) increased CRP); and addition trials (fruit decreased CRP and TNF-α; sweets and desserts (dark chocolate) decreased IL-6). The certainty of evidence was moderate-to-low for the majority of analyses. Conclusions: Food source appears to mediate the effect of fructose-containing sugars on inflammatory markers over the short-to-medium term. The evidence provides good indication that mixed sources that contain SSBs increase CRP, while most other food sources have no effect with some sources (fruit, 100% fruit juice, sweetened soy beverage or dark chocolate) showing decreases, which may be dependent on energy control. Clinicaltrials.gov: (NCT02716870).

## 1. Introduction

Chronic inflammation resulting in elevated levels of inflammatory biomarkers have been associated with a higher risk for the development of cardiovascular disease (CVD), diabetes, and other non-communicable diseases [1]. Randomized controlled trials have shown that agents that decrease inflammation (e.g., canakinumab, colchicine) also decrease CVD risk [2]. Low-grade inflammation can be quantified with acute phase proteins, including *C*-reactive protein (CRP), tumour necrosis factor-alpha (TNF-α), and interleukin-6 (IL-6). Although there are various potential factors that affect low-grade inflammation (e.g., physical activity, smoking, weight loss), diet quality has been show to influence inflammation [3] and relate to the risk of developing CVD [4].

The World Health Organization, among other international organizations, recommend limiting sugars intake, thus sugars remain an important public health focus [5]. There is a particular focus on fructose due to its unique metabolism, and implied contribution towards obesity and the related downstream cardiometabolic implications. Fructose is thought to act as an unregulated substrate for de novo lipogenesis, bypassing negative feedback control, unlike its glucose counterpart. This mechanism is postulated to impair other metabolic signaling and lead to increased adiposity [6,7]. However, the harmful effects of fructose on some cardiometabolic outcomes, including body weight, are only observed when fructose-containing sugars are consumed as excess energy [8,9,10,11,12]. There is some evidence that a diet high in fructose may increase interstitial inflammation, although this comes from animal models [13]. Observational studies have shown that fructose-containing sugar sweetened beverages (SSB) may be associated with increased pro-inflammatory proteins [14,15,16]. Other observational studies have found that dietary patterns play an important role in mediating pro-inflammatory biomarkers where foods high in antioxidants, fiber or long chain-polyunsaturated fatty acids, many of which may be a source of fructose-containing sugars, are associated with lower pro-inflammatory biomarker levels [4].

Whether the evidence linking fructose from SSB to inflammation in humans holds for other commonly consumed food sources of fructose-containing sugars which are sources of anti-inflammatory nutrients/constituents, such as fruit, 100% fruit juice, sweetened cereal grains and sweetened dairy and dairy alternatives, at different levels of energy control is unclear. Therefore, we conducted a systematic review and meta-analysis of controlled trials of the effect of different food sources of fructose-containing sugars at different levels of energy control on biomarkers of inflammation with an assessment of the certainty of evidence using Grading of Recommendations, Assessment, Development, and Evaluation (GRADE).

## 2. Materials and Methods

We followed the Cochrane Handbook for Systematic Reviews of Interventions (version 6.1) [17] for the conduct of our systematic review and meta-analysis and reported our results following the Preferred Reporting Items for Systematic Reviews and Meta-Analyses (PRISMA) guidelines (Appendix A) [18]. The study protocol was registered at ClinicalTrials.gov (NCT02716870).

### 2.1. Data Sources and Search Strategy

We conducted a systematic search in MEDLINE, Embase, and the Cochrane Central Register of Controlled Studies databases through 6 March 2022. Appendix A present the search strategy. There were no language restrictions. Validated filters were applied [19]. The searches were supplemented with manual searches.

### 2.2. Study Selection

We included randomized and non-randomized controlled feeding trials in humans of all health backgrounds and ages, with intervention periods ≥ 7 days [20] investigating the effect of orally consumed fructose-containing sugars from various food sources compared with control diets free of or lower in fructose-containing sugars on CRP, TNF-α, and IL-6. We excluded studies of liquid meal replacement interventions and studies of interventions or comparators of rare sugars that contain fructose (e.g., isomaltulose, melezitose, turanose) or were low calorie epimers of fructose (e.g., allulose, tagatose, sorbose). Reports were initially excluded based on review of their titles and abstracts by a single reviewer. Those reports that remained were then excluded based on review of the full text reports by at least two reviewers (X.Q., L.C., D.L., S.A.-C., F.A.Y., A.A., A.C., Q.L.), leaving the final set of reports to be included in our syntheses. We prespecified four study designs based on energy control: (1) ‘substitution’ trials, in which energy from the food sources of fructose-containing sugars was substituted for other non-fructose-containing macronutrients under energy matched conditions; (2) ‘addition’ trials, in which excess energy from the food sources of fructose-containing sugars was added to the background diet compared to the same diet alone without the excess energy (with or without the use of non-nutritive/low-calorie sweeteners to match sweetness); (3) ‘subtraction’ trials, in which energy from the food sources of fructose-containing sugars was subtracted from background diets compared with the original background diets through displacement by water or low-calorie sweeteners or elimination altogether; and (4) ‘ad libitum’ trials, in which energy from the food sources of fructose-containing sugars was freely replaced (usually within reasonable limits, e.g., intake required to be between 75 and 125% of predicted daily energy requirements) with other non-fructose-containing macronutrients without any strict control of either the study foods or the background diets, allowing for free replacement of energy. In reports containing more than one eligible trial comparison, we included each available trial comparison separately.

### 2.3. Data Extraction

At least two reviewers independently extracted data from eligible studies. Relevant information included food source of fructose-containing sugars, number of participants, setting, participant health status, study design, level of feeding control, randomization, comparator, fructose-containing sugars type, macronutrient profile of the diets, follow-up duration, energy balance, funding source and outcome data. Appendix A shows the definitions for the different food sources of fructose-containing sugars. Authors were contacted for missing outcome data when it was indicated that an inflammatory outcome was measured but not reported. Graphically presented data were extracted from figures using Plot Digitizer [21].

### 2.4. Risk of Bias Assessment

Included studies were assessed for risk of bias independently and in duplicate by ≥2 reviewers using the Cochrane Risk of Bias Tool [17]. Assessment was done across six domains of bias (sequence generation, allocation concealment, blinding, incomplete outcome data, selective outcome reporting and other). Risk of bias for each domain was assessed as either “low” (proper methods taken to reduce bias), “high” (improper methods creating bias) or “unclear” (insufficient information provided). The “other” domain applied only to crossover trials; “high” risk of bias was given when there was no washout between interventions, otherwise the trial was rated as “low”. Reviewer discrepancies were resolved by consensus or arbitration by the senior author (J.L.S.).

### 2.5. Outcomes

The primary outcome was CRP. Secondary outcomes included TNF-α and IL-6. Mean differences (MDs) between the intervention and control arm and their standard errors (SEs) were extracted for each eligible trial comparison. If unavailable, they were derived from available data using published formulas [17]. Mean pairwise difference in change-from-baseline values were preferred over end values. When median data was provided, they were converted to mean data with corresponding variances using methods developed by Luo et al. (2018) [22] and Wan et al. (2014) [23]. When no variance data was available, the standard deviation (SD) was borrowed from a trial similar in size, participants and nature of intervention [24].

### 2.6. Data Syntheses and Analyses

We used Stata software, version 16.1 (StataCorp, College Station, TX, USA) for all analyses. As our primary research question was to assess the effect of different food sources of fructose-containing sugars at different energy control levels, we performed separate pairwise meta-analyses for each of the four prespecified designs by energy control level (substitution, addition, subtraction and ad libitum trials) and assessed the interaction between food sources of fructose-containing sugars within each energy control level using the Cochrane Handbook’s recommended standard Q-test for subgroup differences (significance at *p* < 0.10) [25,26,27].

The principal effect measures were the mean pair-wise differences in change-from-baseline (or alternatively, end differences) between the food sources of fructose-containing sugars arm and the comparator arm (significance at *p* < 0.05). Data were analyzed using the generic inverse variance method with DerSimonian and Laird random-effects model [17,28]. A fixed effects model was used when <5 trial comparisons were available [29]. Paired analyses were applied to all crossover trials with the use of a within-individual correlation coefficient between treatment of 0.5 as described by Elbourne et al. to calculate SEs [30,31,32]. Data were expressed as MDs with 95% confidence intervals (CIs) for all outcomes. To mitigate a unit-of-analysis error, when arms of trials with multiple intervention or control arms were used more than once, the corresponding sample size was divided by the number of times it was used for calculation of the standard error [17].

Heterogeneity was assessed by visual inspection of the forest plots and using the Cochrane Q statistic and quantified using the *I*^2^ statistic [17]. We considered an *I*^2^ ≥ 50% and P_Q_ < 0.10 as evidence of substantial heterogeneity [17]. Sources of heterogeneity were explored by sensitivity analyses, including individual trial influence, altering pairwise comparison correlation coefficient and subgroup analyses. The influence analysis systematically removed each trial comparison from the meta-analysis with recalculation of the summary effect estimate. A trial whose removal explained the heterogeneity or changed the significance, direction, or magnitude of the effect by more than the minimally important difference (MID) for each outcome (0.5 mg/L for CRP [33,34,35], 0.28 pg/mL for TNF-α [36], 0.18 pg/mL for IL-6 [37]) was considered an influential trial. To determine whether the overall results were robust to the use of different correlation coefficients in crossover trials, we also conducted sensitivity analyses using correlation coefficients of 0.25 and 0.75. If ≥10 trials were available [26,38], we conduced subgroup analyses to explore sources of heterogeneity using meta-regression (significance at P_Q_ < 0.05). *A priori* subgroup analyses were conducted by participant health status, age, anti-inflammatory medication use, baseline outcome level, randomization, energy balance, fructose sugars type, comparator, study design, follow-up, feeding control, fructose-containing sugars dose, sugars regulatory designation, funding and risk of bias. Post hoc subgroup analyses were conducted by type of imputation done for deriving variances (data used assessed change from baseline vs. end differences) and type of CRP analysis (CRP vs. high-sensitivity CRP, for CRP analyses). Meta-regression analyses were used to assess the significance of each subgroup categorically and, when applicable, continuously.

If ≥6 trial comparisons were available [39], then we assessed linear and non-linear (restricted cubic splines) dose–response relationships (significance at *p* < 0.05) using meta-regression. We also assessed non-linear dose–response threshold effects with three prespecified spline knots at important public health thresholds of 5% [5,40], 10% [40,41], and 25% [42] total energy (%E).

If ≥10 trials were available, then we assessed publication bias by visual inspection of contour-enhanced funnel plots and formal testing with Egger’s [43] and Begg’s [44] tests (significance at *p* < 0.10) [45]. If there was evidence of publication bias, then we adjusted for funnel plot asymmetry and assessed for small-study effects by imputing the missing trial data using the Duval and Tweedie trim-and-fill method [46].

### 2.7. Certainty of the Evidence

The certainty of the evidence was assessed using the GRADE approach and software (GRADEpro GDT, McMaster University and Evidence Prime Inc., Hamilton, Canada) [47]. The assessments were conducted by two independent reviewers (X.Q., L.C.) and discrepancies were resolved by consensus or arbitration by the senior author (J.L.S.). The evidence was rated as high, moderate, low, or very low certainty. The included controlled trials were initially rated as high certainty by default and then downgraded or upgraded based on pre-specified criteria. Reasons for downgrading the evidence included risk of bias (assessed by the Cochrane Risk of Bias Tool [17]), inconsistency (substantial unexplained inter-study heterogeneity, *I*^2^ > 50% and P_Q_ < 0.10), indirectness (presence of factors that limit the generalizability of the results), imprecision (the 95% CI for effect estimates overlap the MID for benefit or harm), and publication bias (significant evidence of small study effects). The reason for upgrading the evidence was presence of a significant dose–response gradient [48,49,50,51,52,53]. The importance of the magnitude of the pooled estimates were assessed using our prespecified MIDs and the effect size categories according GRADE guidance [47,54,55,56] as follows: large effect (≥5× MID); moderate effect (≥2× MID); small important effect (≥1× MID); and trivial/unimportant effect (<1 MID).

## 3. Results

### 3.1. Search Results

Figure 1 shows the flow of the literature. We retrieved 2850 reports from databases and manual searches, 2698 of which were excluded based on the title or abstract. Of the 152 reports reviewed in full text, 64 reports of controlled feeding trials (91 trial comparisons, *n* = 4094) met the eligibility criteria [57,58,59,60,61,62,63,64,65,66,67,68,69,70,71,72,73,74,75,76,77,78,79,80,81,82,83,84,85,86,87,88,89,90,91,92,93,94,95,96,97,98,99,100,101,102,103,104,105,106,107,108,109,110,111,112,113,114,115,116,117,118,119,120]. These trials included 12 different food sources of fructose-containing sugars (SSB; sweetened dairy; sweetened dairy alternative [soy]; 100% fruit juice; fruit; dried fruit; mixed fruit forms; added nutritive [caloric] sweetener; sweetened cereal grains and bars; sweets and desserts; mixed sources [with SSBs], and mixed sources [without SSBs]) across four energy control levels: substitution (39 trial comparisons); addition (45 trial comparisons); subtraction (4 trial comparisons); and ad libitum (3 trial comparisons). The mixed sources (without SSBs) food category includes those trials in which the intervention included more than one of the food sources, excluding SSBs (e.g., sweets and desserts and fruits).

### 3.2. Trial Characteristics

Table 1 and Appendix A show the trial characteristics. Trial sizes ranged from a median of 15 participants (range 12–120) in subtraction trials to 40 participants (range 12–192) in addition trials. Participants were predominantly adults with and without overweight/obesity, some of whom had a diagnosed chronic condition (e.g., diabetes) or at elevated risk for cardiovascular disease (e.g., dyslipidemia, metabolic syndrome). There were slightly more females in most trial categories with the exception of subtraction trials where there were slightly more males. Most participants were middle-aged adults with ages ranging from a median of 27 (range 26–29) years in subtraction trials to 48 (range 8–72) years in addition trials. Most trials were conducted in an outpatient setting (85–100%), performed in American and European countries, and were parallel in design (62% in substitution and addition, 100% in subtraction, and 67% in ad libitum trials). Feeding control was mostly supplemented for substitution (77%), addition (96%), subtraction (100%), and ad libitum (100%) trials. Most studies were randomized (82–100%). The dose of fructose-containing sugars ranged from a median of 8% (range 1–35%) in addition trials to 19% (range 6–19%) of total energy intake in ad libitum trials. The follow-up duration ranged from a median of 5 weeks in addition trials (range 1–24weeks) to 30 weeks in subtraction trials (range 12–48weeks). The most common source of funding was by agency sources (government, not-for-profit health agency, or university sources) for substitution (41%), addition (45%), and subtraction (50%) trials, with agency and industry sources for ad libitum trials (67%). The comparators for substitution trials were mostly mixed comparator (13/39, 33%) or glucose (12/39, 31%), diet alone for addition trials (31/45, 69%), non-nutritive sweetener for subtraction (3/4, 75%) and mixed for ad libitum trials (2/3, 67%). The main food sources in substitution trials were SSB (10/37, 27%) and mixed sources with (6/37) or without (4/37) SSBs; for addition trials, 100% fruit juice (13/45, 29%), SSB (11/45, 24%) and fruit (9/45, 20%); for subtraction trials, SSB (4/4, 100%); and for ad libitum trials mixed sources (with SSBs) (3/3, 100%).

### 3.3. Risk of Bias

Appendix A show a summary of the risk of bias assessments of the included trials. Across energy designs, 49–69% of trials were assessed as having unclear risk of bias in random sequence generation 50–70% as having unclear allocation concealment domains due to poor reporting, 47–69% were assessed as unclear for incomplete outcome, with 22–53% of trials being assessed as low for blinding (22–53% low) and 53–81% as low for selective outcome reporting. Most cross-over trials were assessed as having low risk of bias in the “other” (carry-over effects) domain (95% in substitution, 89% in addition, 100% in subtraction and ad libitum trials). Fewer studies were assessed as having high risk of bias, for random sequence generation (6–24%), allocation concealment (6–24%), blinding of participants and personnel (0–5%), incomplete outcome data (0%), selective outcome reporting (0–12%), and other (carry-over effects) (0–29%) risk of bias domains. Thus, there was no overall serious risk of bias in most trial comparisons except for in addition trials of sweetened cereal grains and bars for CRP, where there was only one trial that was not randomized, and thus sequence generation and allocation concealment were high risk of bias.

### 3.4. Primary Outcome

Figure 2 and Appendix A present the effect of different food sources of fructose-containing sugars on the primary outcome, CRP, at four levels of energy control. Total fructose-containing sugars resulted in a reduction in CRP for addition trials (37 trials; MD: −0.18mg/L; 95% CI: −0.33, −0.03mg/L, P_MD_ = 0.020; no substantial heterogeneity, *I*^2^ = 43.7%, P_Q_ = 0.003) but no effect in substitution (37 trials; MD: 0.07mg/L; 95% CI: −0.08, 0.22mg/L, P_MD_ = 0.336; substantial heterogeneity, *I*^2^ = 53.7%, P_Q_ < 0.001), subtraction (4 trials; MD: 0.14mg/L; 95% CI: −0.29, 0.56mg/L; P_MD_ = 0.522; no heterogeneity, *I*^2^ = 0.0%, P_Q_ = 0.877), or ad libitum (3 trials; MD: −0.09mg/L; 95% CI: −0.44, 0.25mg/L; P_MD_ = 0.604; no heterogeneity, *I*^2^ = 0.0%, P_Q_ = 0.910) trials.

An interaction by food source was detected in the substitution trials (*p* = 0.010), where sweetened dairy alternative from soy (1 trial; MD: −0.96mg/L; 95% CI: −1.67, −0.25mg/L; P_MD_ = 0.008) and 100% fruit juice (2 trials; MD: −1.09mg/L; 95% CI: −2.01, −0.17mg/L; P_MD_ = 0.021; no heterogeneity, I^2^ = 0.0%, P_Q_ = 0.590) resulted in decreased CRP, while mixed sources (with SSBs) (6 trials; MD: 0.64mg/L; 95% CI: 0.12, 1.17mg/L; P_MD_ = 0.016; substantial heterogeneity, *I*^2^ = 82.9%, P_Q_ < 0.001) increased CRP. No other food sources showed an effect with variable directions of effect. Although the interaction by food source in addition trials was not significant, we assessed an influence by food source as the reduction in CRP was driven by a sole food source: fruit (9 trials; MD: −0.50mg/L; 95% CI: −0.75, −0.25mg/L; P_MD_ < 0.001; no heterogeneity, *I*^2^ = 0.0%, P_Q_ = 0.960). There was no overall effect in subtraction or ad libitum trials and although there was a significant influence of food source since there were only 1 food source in each analysis, neither had any effect.

### 3.5. Secondary Outcomes

Figure 3 and Figure 4 and Appendix A present the effect of different food sources of fructose-containing sugars on our secondary outcomes, TNF-α and IL-6, at four levels of energy control. In substitution trials, there was no overall effect on either outcome, with no significant interaction by food source (*p >* 0.05). In addition trials, there was no overall effect on either outcome. However, there was a significant interaction by food source for IL-6 (*p* = 0.020) where sweets and desserts coming from dark chocolate (1 trial; MD: −8.79pg/mL; 95% CI: −14.26, −3.32pg/mL; P_MD_
*=* 0.002) resulted in a decrease in IL-6. An influence by food source was determined for TNF-α in addition trials since there was a significant reduction for fruit (3 trials; MD: −0.89pg/mL; 95% CI: −1.58, −0.20pg/mL; P_MD_ = 0.012; no substantial heterogeneity, *I*^2^ = 14.3%, P_Q_
*=* 0.311), similar to what was observed for CRP.

### 3.6. Sensitivity and Subgroup Analyses

Appendix A present the influence analyses for the effect of total fructose-containing sugars at the 4 levels of energy control on the primary outcome, CRP. Removal of single trial comparisons provided a partial explanation of the evidence of substantial heterogeneity [81,89] in substitution trials and resulted in a loss of significance for CRP [80] in addition trials.

Appendix A present the influence analyses for the effect of individual food sources, for those analyses that showed evidence of an interaction or influence by food source, on the primary outcome, CRP. Removal of single trial comparisons resulted in loss of significance for the increase in CRP with mixed sources (with SSBs) in substitution trials [81,89] and for the decrease in CRP with 100% fruit juice in substitution trials [109]; a gain of significance for a reduction in CRP with fruit in substitution trials [94] and with sweets and desserts in addition trials [59]; and a partial explanation of heterogeneity for mixed sources (with SSBs) [114] in substitution trials.

Appendix A shows sensitivity analyses for the different correlation coefficients (0.25 and 0.75) used in paired analyses of crossover trials for CRP. The use of these different correlation coefficients did not alter the direction, magnitude, or significance of the effect or evidence for heterogeneity with the following exception: a gain of significance for a reduction in CRP with fruit (MD: −0.43mg/L; 95% CI: −0.85, −0.01mg/L; P_MD_ = 0.045) in substitution trials with the use of 0.75.

Appendix A present the sensitivity analyses for the secondary outcomes. For total fructose-containing sugars, removal of single trial comparisons resulted in a gain of significance for a reduction in TNF-α in addition trials [87,111] and partial explanation of heterogeneity for TNF-α in substitution [84,116] and addition [92] trials. For individual food sources for those analyses that showed evidence of an interaction or influence by food source for secondary outcomes, removal of single trial comparison resulted in: a loss of significance for the reduction in TNF-α with fruit [88] in addition trials; and a gain of significance for a reduction in TNF-α with sweets and desserts (98) and in IL-6 with sweetened dairy [71] and 100% fruit juice [113] in addition trials; and a partial explanation of heterogeneity for SSBs [111] and sweets and desserts [80] on TNF-α and 100% fruit juice [113] on IL-6 in addition trials.

Appendix A shows sensitivity analyses for the different correlation coefficients (0.25 and 0.75) used in paired analyses of crossover trials for secondary outcomes. The use of these different correlation coefficients did not alter the direction, magnitude, or significance of the effect or evidence for heterogeneity for any outcomes across food sources and levels of energy control, with the following exception: gain of significance for a reduction in TNF-α with total fructose-containing sugars with the use of 0.25.

Appendix A present the subgroup analyses and continuous meta-regression analyses for the effect of total fructose-containing sugars, where there were at least 10 trial comparisons, on the primary outcome, CRP. There was significant effect modification by health status (trials of participants with other chronic conditions, such as chronic kidney disease, non-alcoholic fatty liver disease and irritable bowel syndrome, showed increases while trials with other participant types showed no effect), fructose-containing sugars type (trials providing mixed type showed increases while those with fruit showed a tendency for reductions, and others showed no effect in substitution trials), randomization (trials without randomization showed increases while those randomized showed no effect in substitution trials), energy balance (trials with neutral energy balance showed increases while those with positive or negative showed a tendency for reductions in substitution trials), feeding control (metabolic and metabolic with supplementation showed reductions while others showed no effect in substitution and addition trials), other risk of bias (trials with high risk of bias showed increases while those with low showed no effect in substitution trials), and baseline CRP (trials above the median baseline CRP showed reductions while those below the median baseline CRP showed no effect in addition trials).

Appendix A present the subgroup analyses and continuous meta regression analyses for the effect of individual food sources of fructose-containing sugars on the primary outcome, CRP. There was significant effect modification by baseline CRP (trials with baseline CRP greater than the median showed a tendency for increases for SSB in substitution trials, however they showed a tendency for reductions for 100% fruit juice in addition trials), follow up (trials with greater than 8-weeks duration showed a tendency for reductions while those ≥8-weeks showed no effect for 100% fruit juice in addition trials), and selective outcome reporting (low risk of bias trials showed a tendency for reductions for 100% fruit juice in addition trials while unclear risk of bias trials showed no effect).

Appendix A present the subgroup analyses and continuous meta regression analyses for the effect of total fructose-containing sugars, where there were at least 10 trial comparisons, on secondary outcomes. There was significant effect modification involving both TNF-α and IL-6 by incomplete outcome (trials with low risk of bias showed tendency for reductions for TNF-α, yet increases for IL-6) in substitution trials, and randomization (trials without randomization tended to show reductions, while those randomized showed no effect) and design (crossover trials showed reductions while parallel trials showed no effect) in addition trials. A few of other subgroup analyses showed subgroup differences for individual outcomes across levels of energy control, without any discernable pattern. There were no subgroup analyses for the effect of individual food sources on secondary outcomes as there was no interaction or influence by food source or there were <10 trial comparisons available.

### 3.7. Dose Response Analyses

Appendix A present linear and non-linear dose–response analyses for the primary outcome, CRP. In substitution trials, there was no dose response for the effect of total fructose-containing sugars nor for any food source with ≥6 trial comparisons. In addition trials, there was a non-linear dose response for total fructose-containing sugars (P_non-linear_ < 0.001) and dose threshold relationships at 5% (*p* = 0.001) and 10% (*p* = 0.002). There was also a dose threshold relationship at 5% for 100% fruit juice (*p* = 0.043). In subtraction and ad libitum trials, there were too few trials to assess dose responses for total fructose-containing sugars or any food source.

Appendix A present linear and non-linear dose–response analyses for secondary outcomes. There was a dose threshold relationship at 5% for TNF-α in addition trials (*p* = 0.002) where greater reductions are seen with lower doses. There was also a dose threshold relationship at 5% for IL-6 in substitution trials (*p* = 0.025), where the increase at low doses was driven by only 1 study (2 trial comparisons) [94]. Although this study found increases in IL-6, they found reductions for TNF-α. There were too few trials with dose data to assess dose responses for secondary outcomes by food source.

### 3.8. Publication Bias

Appendix A present the publication bias and trim-and-fill (where applicable) assessments for all outcomes where there were ≥10 trials available. There was no evidence of funnel plot asymmetry in any analysis of the primary outcome, CRP. There was evidence of funnel plot asymmetry for the effect of total fructose-containing sugars on IL-6 in substitution (Egger’s test, *p* = 0.004) and addition (Egger’s test, *p* = 0.015) trials. Adjustment for funnel plot asymmetry with the imputation of 4 missing trials by the Duval and Tweedie trim-and-fill method, however, did not alter the direction, magnitude or significance of the effect, suggesting that there was no meaningful influence of publication bias on the results (Original MD in substitution trials: −0.04pg/mL; 95% CI: −0.24 to 0.15, *p* = 0.664; imputed MD: −0.05; 95% CI: −0.28 to 0.18, *p* = 0.677; Original MD in addition trials: −0.15pg/mL; 95% CI: −0.45 to 0.16, *p* = 0.349; imputed MD: −0.06pg/mL; 95% CI: −0.39 to 0.27, *p* = 0.718).

### 3.9. GRADE Assessment

Figure 2, Figure 3 and Figure 4 and Appendix A present the GRADE assessments. The certainty of evidence for the effect of total fructose-containing sugars on the primary outcome, CRP, was low in substitution (no effect), addition (trivial reduction), and ad libitum (no effect) trials and very low for subtraction trials (no effect), owing to double downgrades for indirectness across the 4 levels of energy control and a single downgrade for imprecision in subtraction trials.

As there was evidence of significant interaction or influence by food source, the certainty of evidence was assessed for the individual food sources. The certainty of evidence was low for sweetened dairy alternative (soy) (small important reduction) and 100% fruit juice (moderate reduction), and moderate for mixed sources (with SSBs) (small important increase) in substitution trials owing to downgrades for indirectness and/or imprecision; and moderate for fruit (small important reduction) in addition trials owing to a downgrade for imprecision. The certainty of evidence for the remaining food sources which showed no effect, was generally moderate, ranging from high to very low, owing to downgrades for risk of bias, inconsistency, indirectness, and/or imprecision.

The certainty of evidence for the effect of total fructose-containing sugars on secondary outcomes was high for TNF-α and moderate for IL-6 in substitution trials, due to a downgrade for imprecision, and very low for both outcomes in addition trials owing to double downgrades for indirectness and at least one single downgrade for inconsistency and/or imprecision.

As there was evidence of influence by food source in addition trials for TNF-α and IL-6, the certainty of evidence was assessed for individual food sources. The certainty of evidence was moderate for the effect of fruit on TNF-α (small important reduction) and sweets and desserts (dark chocolate) on IL-6 (large reductions) owing to a downgrade for imprecision and indirectness, respectively. The certainty of evidence for the remaining food sources which showed no effect, was moderate or low, owing to downgrades for inconsistency, indirectness, and/or imprecision.

## 4. Discussion

We conducted a systematic review and meta-analysis of 64 reports (91 trial comparisons) in 4094 generally healthy participants with or without obesity, with few trials of participants who have or are at risk for cardiometabolic diseases, of the effects of 12 different food sources of fructose-containing sugars (SSB; sweetened dairy; sweetened dairy alternative [soy]; 100% fruit juice; fruit; dried fruit; mixed fruit forms; sweetened cereal grains and bars; sweets and desserts; added nutritive [caloric] sweetener; mixed sources [with SSBs]; and mixed sources [without SSBs]) with a median dose of 8% to 19% of total energy across four different levels of energy control over median follow-up of 5–30 weeks. Total fructose-containing sugars led to a trivial reduction in CRP (−0.18 mg/L) in addition trials. There was no effect of total fructose-containing sugars at the other levels of energy control or on secondary outcomes. There was evidence of interaction or influence by food source in most analyses. In substitution trials, sweetened dairy alternatives as a soy beverage at a dose of 1%E (5 g sugar) led to small important reductions in CRP (−0.96 mg/L) and 100% fruit juice at doses of 8.8%E and 12%E led to a moderate reduction in CRP (−1.09 mg/L), while mixed sources (with SSBs) at a median dose of 6.4%E (ranging from 6.3%E to 27%E) led to a small important increase in CRP (0.64 mg/L). In addition trials, fruit at a median dose of 3.8%E (ranging from 1.6%E to 10%E) led to a small important reduction in CRP (−0.50 mg/L) and TNF-α (−0.89 pg/mL), while sweets and desserts as dark chocolate at a dose of 1.1%E led to a large reduction in IL-6 (−8.79 pg/mL). Other food sources of fructose-containing sugars showed no effect on markers of inflammation.

### 4.1. Findings in Relation to the Literature

Our results for total fructose-containing sugars are similar to a previous systematic review and meta-analysis of the effects of fructose-containing sugars on CRP which included 6 controlled trials (*n* = 403) and found no significant difference between fructose and glucose interventions (MD: −0.03 mg/L; 95% CI: −0.52, 0.46 mg/L; *I*^2^ = 44%) [121]. The present analyses build on the previous study, as it identified many more reports, including on additional inflammatory biomarkers (TNF-α, IL-6), prespecified 4 energy designs in order to separate the effect of energy control, and explored the interaction between food sources of fructose-containing sugars.

The benefits or lack of harm observed for certain food sources of fructose-containing sugars is in agreement with previous observations. The reduction in CRP and TNF-α observed in addition trials for fruit at a median intake of 3.8%E (range of 1.6%E to 10%E), where the predominant type was berries, is supported by a systematic review and meta-analysis of controlled trials which showed similar reductions in inflammation (TNF-α, MD: −0.99 pg/mL; 95% CI: −1.96, −0.02 pg/mL; *p* = 0.04), as well as reductions in adiposity, glycemic control, blood lipids, and blood pressure [122]. Inflammatory benefits have also been observed in a recent systematic review and meta-analysis of fruit and vegetable intake where 10 observational studies found an inverse association between intakes of fruit or vegetables and inflammatory biomarkers and 71 clinical trials showed significant reductions in both CRP (23 trial comparisons, MD: −0.34 mg/L; 95% CI: −0.58, −0.11 mg/L; *p* < 0.01) and TNF-α (16 trial comparisons, MD: −0.87 pg/mL; 95% CI: −1.59, −0.15 pg/mL; *p* = 0.02) [123]. Improvements in cardiovascular risk factors were demonstrated for fruit in our systematic reviews and meta-analyses of the effect of food sources of fructose-containing sugars on adiposity, blood pressure, and glycemic control [11,124,125].

The reduction in CRP by 100% fruit juice, specifically 100% orange juice, at doses of 9%E and 12%E, in substitution trials is in agreement with improvements in inflammatory markers observed in a systematic reviews and meta-analyses of clinical trials on 100% orange juice (IL-6: 5 trial comparisons, MD: −1.51 pg/mL; 95% CI: −2.31, −0.70pg/mL; hs-CRP; 9 trial comparisons, MD: −0.58 mg/L; 95% CI: −1.22, 0.05 mg/L) [126]. This result is also supported by systematic reviews and meta-analyses of prospective cohort studies which have demonstrated U-shaped associations between 100% fruit juice intake and cardiometabolic outcomes such as incident hypertension [127], metabolic syndrome [128], and cardiovascular event risk [129] and our recent systematic review and meta-analysis of fructose-containing sugars showing improvements in markers of adiposity at doses ≤10%E [124]. Evidence in the literature generally shows improvement in risk factors at low to moderate doses of 100% fruit juice [130].

The reduction in CRP in the substitution trial of a sweetened soy beverage, which included participants with non-alcoholic fatty liver disease, is supported by systematic reviews and meta-analyses demonstrating improvements in inflammatory biomarkers from the consumption of soy [131,132], including one specifically showing benefit of soy protein [133]. A subgroup analysis in one of these studies demonstrated the reduction was stronger in participants who were affected various chronic diseases [131]. The sweetened soy beverage reduction in CRP observed in the present study is also reflected in the CRP reduction observed with the dietary portfolio, a cholesterol-lowering dietary pattern that involves a relatively high soy milk consumption [134,135]. Therefore, the CRP reduction in sweetened soy beverage may be generalizable to those at higher risk of or with a chronic disease.

The reduction in IL-6 observed in the one addition trial of dark chocolate, which provided 1.1%E (5 g sugar/day) as 84% dark chocolate, is supported by systematic reviews and meta-analyses showing associations between chocolate intake and lower risk of CVD incidence and mortality [136].

In substitution trials with mixed sources with SSBs there was an increase in CRP. For these trials, in the comparator arm, there was a specific focus on restricting SSBs and added sugars in the diet, replacing them predominantly with starch. It is possible that this resulted in an increase in whole grain and dietary fibre intake on the comparator, both of which have been demonstrated to reduce inflammatory markers in systematic reviews and meta-analyses of controlled trials, particularly in participants with chronic diseases [137,138], which was the participant type in all of the included trials. The lack of harm observed for SSBs alone at all levels of energy control is supported by a previous systematic review and meta-analysis including 7 trial comparisons of fructose versus glucose which showed no overall effect [121]. Previous systematic reviews and meta-analyses exploring the effect of different food sources of fructose containing sugars on cardiometabolic outcomes in controlled trials have demonstrated harm when SSBs are consumed as a source of excess calories, including on glycemic control, adiposity, blood pressure and uric acid [8,11,139,140]. It is possible that the lack of harm observed in the present analysis may be the result of fewer trials and those which only comprised of healthy participants free of chronic diseases and with low baseline CRP levels (median 0.4 mg/L, range 0.2–1.22 mg/L).

Baseline CRP level may be an important consideration since it was the only factor that was significant in subgroup analyses for the effect of SSB on CRP in substitution trials and of 100% fruit juice on CRP in addition trials; the only 2 food sources in all energy designs where there were ≥10 trial comparisons allowing for subgroup analyses to be performed. In categorical subgroup analyses of substitution trials, SSB showed a tendency to increase CRP when baseline CRP was greater than the median. There was also a positive continuous relationship where SSB showed a greater effect on CRP with greater baseline CRP. Conversely, 100% fruit juice tended to reduce CRP to a greater extent when baseline CRP was greater than the median. This effect is supported by the significant reduction in CRP found with 100% fruit juice in substitution trials in which the 2 trials included had higher baseline CRP levels. Thus, the potential effects of difference food sources of fructose-containing sugars may be more prominent in populations with higher baseline inflammation.

### 4.2. Potential Mechanisms

These advantages seen for some foods may be partly explained by the food’s content of antioxidants, flavonoids, and/or polyphenols. Fruit, especially berries and apples, which were the predominant source in the included trials, as well as oranges in 100% orange juice, are rich sources of antioxidants, while soy milk is a source of isoflavones and dark chocolate is a rich source of flavonoids, all of which have evidence to support an explanation for cardiovascular improvements [126,141,142]. Conversely, some of the food sources of fructose-containing sugars which showed no effect (e.g., SSBs, sweetened dairy, sweets and desserts, sweetened cereal grains and bars, added nutritive sweeteners) would be expected to be lower in or devoid of these bioactives. Dried fruit would be expected to have similar level of bioactives as fruit, as indicated in the few included trials of dried fruit in the present analysis, however they showed no overall effect. Our similar systematic review and meta-analysis on markers of adiposity included more controlled trials and showed improvements in body weight and BMI for dried fruit [124], thus it remains uncertain whether additional trial data may affect the conclusion on inflammation. Bioactives, including antioxidants and flavonoids, may also interfere with fructose metabolism. For example, antioxidants and flavonoids may reduce oxidative stress and thus fructose-induced uric acid production [143], which is supported by the lack of harm observed for fruit and fruit juice in a systematic review and meta-analysis of food sources of fructose-containing sugars on uric acid which contrasts the significant increases in uric acid that were observed for SSBs [8]. In addition to bioactives as mechanisms through which various foods may influence inflammation, these food sources of fructose-containing sugars can be higher in dietary fibre and lower in glycemic index (GI). Fruit which showed reductions in markers of inflammation have higher fibre (e.g., apples 4 g/medium, berries 4 g/cup,) and lower GI (e.g., apples 38, berries 28) [144], and soy beverages, orange juice and dark chocolate are low GI foods [144], whereas the SSBs, added nutritive sweeteners, and mixed sources (with SSBs) would be expected to be lower in fibre and higher in GI. Low GI diets may improve inflammation as demonstrated in a recent systematic review and meta-analysis of low glycemic index/load diets which showed similar reductions in CRP resulting from low GI compared to higher GI diets [145]. Circulating insulin and related incretin hormones may be reduced, increasing satiety and decreasing subsequent energy intake with the consumption of lower GI and higher fibre foods [146,147,148,149,150].

### 4.3. Strengths and Limitations

Our systematic review and meta-analysis has several strengths. First, we conducted a comprehensive and reproducible search and selection process of the literature examining the effect of food sources of fructose-containing sugars on markers of inflammation. Second, we collated and synthesized the totality of available evidence from a large body (64 reports, 89 trial comparisons, *n* = 3958) of controlled intervention studies, which give the greatest protection against bias. Third, we had comprehensive exploration of possible sources of heterogeneity. Fourth, we evaluated the shape and strength of the dose–response relationships. Fifth, we assessed the overall quality of evidence using the GRADE assessment approach.

Our analyses also presented limitations. First, there was evidence for serious risk of bias in one analysis of sweetened cereal grains and bars on CRP in addition trials due to the lack of randomization of the one trial resulting in high risk of bias for sequence generation and allocation concealment. Second, there was evidence of indirectness. The significant interaction or influence of food source in substitution trials for CRP and addition trials for all outcomes and the limited number of food sources of fructose-containing sugars available in the subtraction and ad libitum trials for CRP (only one or two food sources available [SSB and/or mixed sources with SSBs]) in the pooled analyses for total fructose-containing sugars meant the results could not be generalized to all food sources. We therefore double downgraded for very serious indirectness in these analyses and rated the evidence separately for individual food sources. The downgrades for indirectness of individual food sources were related to insufficient trial comparisons which limited generalizability related to population, intervention or comparator. The absence of long-term trials (>1-year diet duration), might be another reason to downgrade for serious indirectness, however we concluded based on short term intake. Third, there was evidence of inconsistency in a few of the pooled estimates. In the addition trial analyses for IL-6 of total fructose-containing sugars, for CRP of sweets and desserts, and for TNF-α of 100% fruit juice, we downgraded for serious inconsistency due to substantial unexplained heterogeneity. Finally, there was evidence of imprecision in almost all of the pooled analyses. We downgraded for serious imprecision due to the crossing of the prespecified MID, which meant that clinically important benefit and/or harm could not be ruled out.

Weighing the strengths and limitations, the certainty of evidence was low for the decreasing effect of sweetened dairy alternative as soy and 100% fruit juice and moderate for the increasing effect of mixed sources (with SSBs) on CRP in substitution trials, moderate for the decreasing effect of fruit on CRP and TNF-α and sweets and desserts as dark chocolate on IL-6 in addition trials, and generally low (very low to high) for the effect of all other comparisons on markers of inflammation.

### 4.4. Implications

Our findings, similar to our previous systematic reviews and meta-analyses on the importance of food sources of fructose-containing sugars on other cardiometabolic outcomes [8,11,124,125,139], suggest that the focus of dietary guidelines [151] should be on dietary patterns, recognizing the importance and complex interactions of the food matrix and the energy conditions under which foods are consumed, rather than limited to single nutrients, like total fructose-containing sugars. Our present results demonstrating the benefit of fruit on inflammation are also supported by the Global Burden of Disease Study which showed the most important contributors to the global burden of morbidity and mortality are foods we should increase intake of, include increased intake of fruit [152]. Fruits are an important source of dietary fibre and key nutrients, such as potassium, both of which are heavily under-consumed in many populations [153,154]. Thus, a focus on policies encouraging the intake of important foods like fruit can help improve nutrient intakes in the general population in addition to improving health outcomes. In addition to fruit, the present evidence to support sweetened dairy alternatives, specifically soy beverages, and limiting sugars coming from mixed sources, including sugar-sweetened beverages, are supported by obesity, diabetes and cardiovascular guidelines which recommend following plant-based dietary patterns (Mediterranean, vegetarian, Portfolio, dietary approaches to stop hypertension (DASH), low-GI dietary patterns), which encourage food sources of fructose-containing sugars like fruits, vegetables, and whole grains, and reducing intake of others like sugar-sweetened beverages [155,156,157,158,159,160].

## 5. Conclusions

Overall food source appears to mediate the effect of fructose-containing sugars on inflammatory markers in predominantly adults with or without obesity, some of whom have or are at risk for cardiometabolic diseases over the short-to-medium term. The evidence provides good indication that mixed sources that contain SSBs increase CRP, while most other food sources have no effect with some sources (fruit, 100% fruit juice, sweetened soy beverage or dark chocolate) showing decreases, which may be dependent on energy control. The main sources of uncertainty across the analyses were imprecision and indirectness with a particular lack of food sources assessed and data available for subtraction and ad libitum trials. Although there remains a need for larger, longer, high-quality randomized trials assessing a broader variety of food sources of fructose-containing sugars on inflammatory biomarkers, clinical practice guidelines should consider the role of food source of fructose-containing sugars in the management of inflammation.

## Figures and Tables

**Figure 1 nutrients-14-03986-f001:**
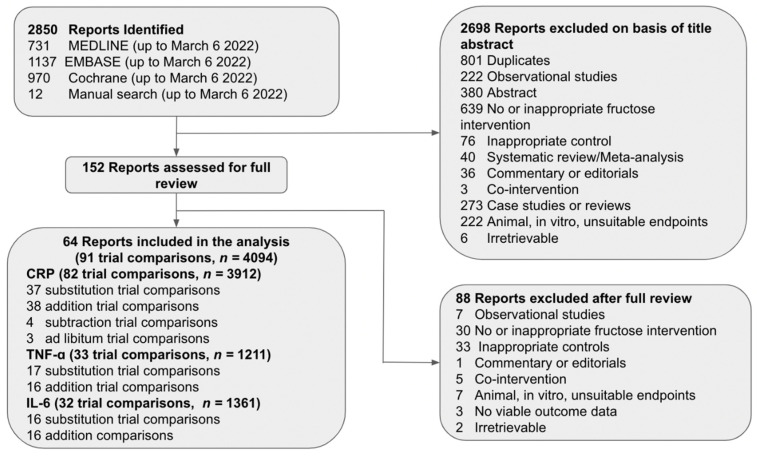
Flow of literature for the effect of food sources of fructose-containing sugars and inflammation; CRP = *C*-reactive protein; IL-6 = interleukin-6; TNF-α = tumor necrosis factor-alpha.

**Figure 2 nutrients-14-03986-f002:**
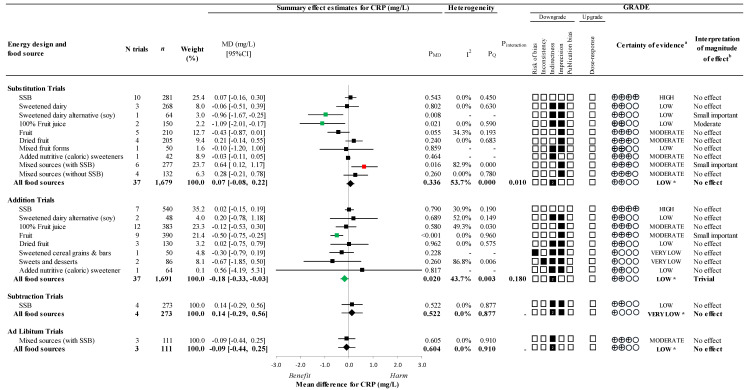
Summary plot for the effect of different food sources of fructose-containing sugars on CRP; Data are weighted mean differences (95% confidence intervals) for summary effects of individual food sources and total food sources on CRP. Analyses conducted by generic, inverse variance random effects models (at least five trials available) or fixed effects models (fewer than five trials available). Between-study heterogeneity was assessed by the Cochrane Q statistic, where P_Q_ < 0.100 is considered statistically significant, and quantified by the *I*^2^ statistic, where *I*^2^ ≥ 50% is considered evidence of substantial heterogeneity. The effects of total fructose-containing sugars are denoted by the bolded lines with the effect estimates as diamonds. The effects of individual food sources are denoted by the non-bolded lines with the effect estimates as squares. Any statistically significant reductions are highlighted in green and significant increases in red. The Grading of Recommendations, Assessment, Development and Evaluation (GRADE) of randomized controlled trials are rated as “High” certainty of evidence and can be downgraded by five domains and upgraded by one domain. The white squares represent no downgrades, while filled black squares indicate a single downgrade or upgrades for each outcome, and the black square with a white “2” indicates a double downgrade for each outcome. CI = confidence interval; GRADE = Grading of Recommendations, Assessment, Development and Evaluation; MD= mean difference; N = number; SSB = sugar-sweetened beverages; ^a^ Since all included trials were randomized or non-randomized controlled trials, the certainty of the evidence was graded as high for all outcomes by default and then downgraded or upgraded based on pre-specified criteria. Criteria for downgrades included risk of bias (ROB) (downgraded if the majority of trials were considered to be at high ROB); inconsistency (downgraded if there was substantial unexplained heterogeneity [*I*^2^ ≥ 50%, P_Q_ < 0.10]; indirectness (downgraded if there were factors absent or present relating to the participants, interventions, or outcomes that limited the generalizability of the results); imprecision (downgraded if the 95% confidence interval crossed the minimally important difference [MID] for harm or benefit set 0.5 mg/L for CRP [33,34,35]; and publication bias (downgraded if there is evidence of publication bias based on funnel plot asymmetry and/or significant Egger’s or Begg’s tests (*p* < 0.10) with confirmation by adjustment by Duval and Tweedie trim-and-fill analysis). Criteria for upgrades included a significant dose–response gradient; ^b^ For the interpretation of the magnitude, we used the MIDs (see a above) to assess the importance of magnitude of our point estimate using the effect size categories according to new GRADE guidance. We then used the MIDs to assess the importance of the magnitude of our point estimates using the effect size categories according GRADE guidance [47,54,56] as follows: large effect (≥5× MID); moderate effect (≥2× MID); small important effect (≥1× MID); and trivial/unimportant effect (<1 MID); * Where there was a significant interaction by food source in substitution trials, an influence of fruit in addition trials, and SSBs and/or mixed sources (with SSBs) were the sole food sources in subtraction and ad libitum trials, we performed the GRADE analysis for each individual food source.

**Figure 3 nutrients-14-03986-f003:**
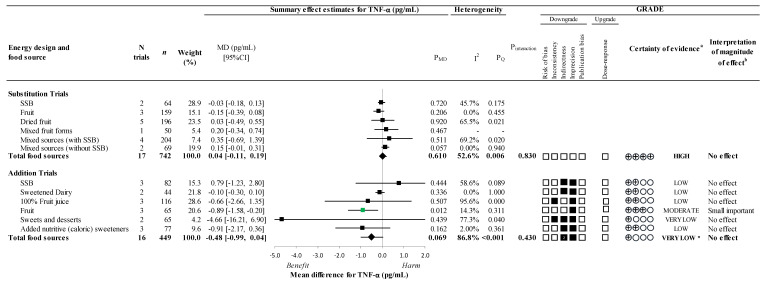
Summary plot for the effect of different food sources of fructose-containing sugars on TNF-α; Data are weighted mean differences (95% confidence intervals) for summary effects of individual food sources and total food sources on TNF-α. Analyses conducted by generic, inverse variance random effects models (at least five trials available) or fixed effects models (fewer than five trials available). Between-study heterogeneity was assessed by the Cochrane Q statistic, where P_Q_ < 0.100 is considered statistically significant, and quantified by the *I*^2^ statistic, where *I*^2^ ≥ 50% is considered evidence of substantial heterogeneity. The effects of total fructose-containing sugars are denoted by the bolded lines with the effect estimates as diamonds. The effects of individual food sources are denoted by the non-bolded lines with the effect estimates as squares. Any statistically significant reductions are highlighted in green and significant increases in red. The Grading of Recommendations, Assessment, Development and Evaluation (GRADE) of randomized controlled trials are rated as “High” certainty of evidence and can be downgraded by five domains and upgraded by one domain. The white squares represent no downgrades, while filled black squares indicate a single downgrade or upgrades for each outcome, and the black square with a white “2” indicates a double downgrade for each outcome. CI = confidence interval; GRADE = Grading of Recommendations, Assessment, Development and Evaluation; MD= mean difference; N = number; SSB = sugar-sweetened beverages; TNF-α = tumour necrosis factor-alpha; ^a^ Since all included trials were randomized or non-randomized controlled trials, the certainty of the evidence was graded as high for all outcomes by default and then downgraded or upgraded based on pre-specified criteria. Criteria for downgrades included risk of bias (ROB) (downgraded if the majority of trials were considered to be at high ROB); inconsistency (downgraded if there was substantial unexplained heterogeneity [*I*^2^ ≥ 50%, P_Q_ < 0.10]; indirectness (downgraded if there were factors absent or present relating to the participants, interventions, or outcomes that limited the generalizability of the results); imprecision (downgraded if the 95% confidence interval crossed the minimally important difference [MID] for harm or benefit set at 0.28 pg/mL for TNF-α [36]; and publication bias (downgraded if there is evidence of publication bias based on funnel plot asymmetry and/or significant Egger’s or Begg’s tests (*p* < 0.10) with confirmation by adjustment by Duval and Tweedie trim-and-fill analysis). Criteria for upgrades included a significant dose–response gradient; ^b^ For the interpretation of the magnitude, we used the MIDs (see a above) to assess the importance of magnitude of our point estimate using the effect size categories according to new GRADE guidance. We then used the MIDs to assess the importance of the magnitude of our point estimates using the effect size categories according GRADE guidance [47,54,56] as follows: large effect (≥5× MID); moderate effect (≥2× MID); small important effect (≥1× MID); and trivial/unimportant effect (<1 MID); * Where there was an influence of fruit in addition trials, we performed the GRADE analysis for each individual food source.

**Figure 4 nutrients-14-03986-f004:**
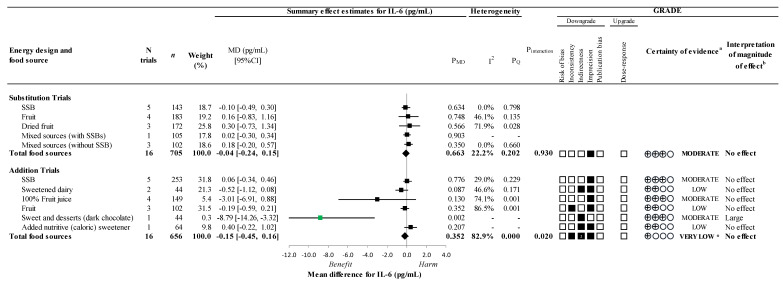
Summary plot for the effect of different food sources of fructose-containing sugars on IL-6; Data are weighted mean differences (95% confidence intervals) for summary effects of individual food sources and total food sources on IL-6. Analyses conducted by generic, inverse variance random effects models (at least five trials available) or fixed effects models (fewer than five trials available). Between-study heterogeneity was assessed by the Cochrane Q statistic, where P_Q_ < 0.100 is considered statistically significant, and quantified by the *I*^2^ statistic, where *I*^2^ ≥ 50% is considered evidence of substantial heterogeneity. The effects of total fructose-containing sugars are denoted by the bolded lines with the effect estimates as diamonds. The effects of individual food sources are denoted by the non-bolded lines with the effect estimates as squares. Any statistically significant reductions are highlighted in green and significant increases in red. The Grading of Recommendations, Assessment, Development and Evaluation (GRADE) of randomized controlled trials are rated as “High” certainty of evidence and can be downgraded by five domains and upgraded by one domain. The white squares represent no downgrades, while filled black squares indicate a single downgrade or upgrades for each outcome, and the black square with a white “2” indicates a double downgrade for each outcome. CI = confidence interval; GRADE = Grading of Recommendations, Assessment, Development and Evaluation; MD = mean difference; N = number; SSB = sugar-sweetened beverages; ^a^ Since all included trials were randomized or non-randomized controlled trials, the certainty of the evidence was graded as high for all outcomes by default and then downgraded or upgraded based on pre-specified criteria. Criteria for downgrades included risk of bias (ROB) (downgraded if the majority of trials were considered to be at high ROB); inconsistency (downgraded if there was substantial unexplained heterogeneity [*I*^2^ ≥ 50%, P_Q_ < 0.10]; indirectness (downgraded if there were factors absent or present relating to the participants, interventions, or outcomes that limited the generalizability of the results); imprecision (downgraded if the 95% confidence interval crossed the minimally important difference [MID] for harm or benefit set at 0.18 pg/mL for IL-6 [37]; and publication bias (downgraded if there is evidence of publication bias based on funnel plot asymmetry and/or significant Egger’s or Begg’s tests (*p* < 0.10) with confirmation by adjustment by Duval and Tweedie trim-and-fill analysis). Criteria for upgrades included a significant dose–response gradient; ^b^ For the interpretation of the magnitude, we used the MIDs (see a above) to assess the importance of magnitude of our point estimate using the effect size categories according to new GRADE guidance. We then used the MIDs to assess the importance of the magnitude of our point estimates using the effect size categories according GRADE guidance [47,54,56] as follows: large effect (≥5× MID); moderate effect (≥2× MID); small important effect (≥1× MID); and trivial/unimportant effect (<1 MID); * Where there was a significant interaction by food source in addition trials, we performed the GRADE analysis for each individual food source.

**Table 1 nutrients-14-03986-t001:** Summary of trial characteristics ^a^.

Trial Characteristics	Substitution Trials	Addition Trials	Subtraction Trials	Ad libitum Trials
**Trials (N)**	39	45	4	3
**Participants (median *n* (range))**	38 (21–267)	40 (12–192)	15 (12–120)	40 (29–50)
**Underlying disease status** **(N trials)**	healthy mixed weight = 13, overweight or obese = 11, type 2 diabetes mellitus = 4, metabolic syndrome = 3, other = 8	healthy mixed weight = 17, overweight or obese = 8, type 2 diabetes mellitus = 3, metabolic syndrome = 2, other = 15	healthy mixed weight = 2, overweight or obese = 2	healthy normal weight = 1, overweight or obese = 2
**Age (median years (range)) ^b^**	46 (14–70)	48 (8–72)	27 (26–29)	38 (32–39)
**Sex ratio (% Male:Female)**	36:64	42:58	60:40	38:62
**Randomization (%)**	90	82	100	100
**Setting ratio** **(% N *=* IP:OP:IP + OP)**	0:97:3	0:100:0	0:100:0	0:100:0
**Country (N trials)**	USA = 14, Iran = 5, Finland = 4, Brazil = 3, Greece = 3, Switzerland = 3, Sweden = 2, UK = 2, Poland = 2, Netherlands = 1	USA = 10, Denmark = 6, Iran = 5, Spain = 4, Switzerland = 3, Thailand = 3, Brazil = 3, India = 2, Italy = 1, Canada = 2, Mexico = 2, Malaysia = 1, Norway = 1, Israel= 1, UK = 1	USA = 2, Switzerland = 2	Netherlands = 2, UK = 1
**Baseline CRP (median mg/L (range)) ^c^**	2.2 (0.2–8.1)	1.5 (0.2–55.5)	2.2 (0.9–3.5)	3.0 (1.0–3)
**Baseline TNF-** **ɑ (median pg/mL (range)) ^d^**	2.4 (1–6.8)	5.4 (1.2–29.2)	Not reported	Not reported
**Baseline IL-6 (median pg/mL (range)) ^e^**	2.0 (0.8–27.4)	3.1 (0.6–16.4)	Not reported	Not reported
**Fructose-containing sugars dose (median %E (range))**	9 (1–45)	8 (1–35)	15 (15–15)	19 (6–19)
**Study design** **(%; crossover:parallel)**	38:62	38:62	0:100	33:67
**Feeding control (%; met:supp:DA:met,supp:supp,DA)**	2.5:77:2.5:18	0:96:2:2	0:100:0:0	0:100:0:0
**Follow-up duration** **(median weeks (range))**	6 (1–24)	5 (1–24)	30 (12–48)	24 (8–24)
**Fructose-containing sugars type (N trials)**	fructose = 8, sucrose = 6, honey =1, fruit = 14, HFCS = 3, mixed type = 7	fructose = 3, sucrose = 13, honey = 3, fruit = 25, mixed type = 1	sucrose = 2, HFCS = 2	sucrose = 1, mixed type = 2
**Comparator (N trials)**	mixed = 13, glucose = 12, starch = 4, fat = 4, lactose = 3, maltodextrin = 2, protein= 1	diet alone= 31, non-nutritive sweetener = 5, other = 5, water = 4	non-nutritive sweetener = 3, water = 1	mixed = 2, non-nutritive sweetener = 1
**Food sources of fructose-containing sugars (N trials)**	SSB = 10, sweetened dairy = 3, sweetened dairy alternative (soy) = 1, 100% fruit juice = 2, fruit = 6, dried fruit = 5, mixed fruit forms = 1, added nutritive (caloric) sweeteners = 1, mixed sources (with SSBs) = 6, mixed sources (without SSBs) = 4	SSB = 11, sweetened dairy = 2, 100% fruit juice = 13, fruit = 9, dried fruit = 3, sweetened cereal grains and bars = 1, sweets and desserts = 3, added nutritive sweeteners = 3	SSB = 4	mixed sources = 3
**Funding sources ratio** **(% *n =* A:I:A,I:NR)**	41:23:33:3	45:11:39:5	50:0:50:0	67:33:0:0

A = agency, A,I = agency and industry, CRP = *C*-reactive protein, E = Energy, HFCS = high fructose corn syrup, I = industry, IL-6 = interleukin 6, IP = inpatient, NR = not reported, OP = outpatient, SSB = sugar sweetened beverages, TNF-α = tumour necrosis factor alpha, N = number, UK = United Kingdom, USA = United States of America; ^a^ Values are rounded to nearest whole number except for baseline outcomes. ^b^ Based on trials which report data. ^c^ Based on trial comparisons that reported baseline data (N = 4 trials missing baseline CRP substitution trials and N = 3 trials missing baseline CRP addition trials). ^d^ Based on trial comparisons that reported baseline data (N = 3 trials missing baseline TNF-α substitution trials; N = 3 trials missing for baseline TNF-α addition trials). ^e^ Based on trial comparisons that reported baseline data (N = 3 trials missing baseline IL-6 substitution trials; N = 2 trials missing baseline IL-6 addition trials).

## Data Availability

Data available upon request.

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
