# Peer review of "Effect of Important Food Sources of Fructose-Containing Sugars on Inflammatory Biomarkers: A Systematic Review and Meta-Analysis of Controlled Feeding Trials"

_nutrients, 2022, doi:10.3390/nu14193986_

Round 1
Reviewer 1 Report
Comments to the Authors of manuscript number: nutrients-1922621 entitled “Effect of important food sources of fructose-containing sugars on inflammatory biomarkers: a systematic review and meta-analysis of controlled feeding trials”.
It is a meta-analysis relating to fructose-containing sugar and some pro-inflammatory factors. It is very well written and consistent a systematic review. Congratulations.
1. L 71- what is inflammatory proteins?
2. Introduction is well written, consistent.
3. L 119 – the sources were properly chosen
4. Study design selection is well described
5. Authors have presented the definition of fructose-containing sugar
6. Outcomes and data analysis were described separately
7. there was even presented the certainty of the evidence
8. L 192 – particular trials are described in details.
9. Parts 2 and 3 are very informative in many details, supported by figures and table, all is enriched by supplementary material.
10. The large supplementary material is prepared carefully. It took time.
11. Discussion is written in reasonable manner. Taken into account the quantity of the material used, it is very consistent.
12. in general, the authors deserve congratulations for such a good job and the time put into its preparation.
Author Response
Please find below the Reviewer comments where we have responded to each comment as indicated in italicized font.
Reviewer #1:
Comments to the Authors of manuscript number: nutrients-1922621 entitled “Effect of important food sources of fructose-containing sugars on inflammatory biomarkers: a systematic review and meta-analysis of controlled feeding trials”.
It is a meta-analysis relating to fructose-containing sugar and some pro-inflammatory factors. It is very well written and consistent a systematic review. Congratulations.
Thank you for your supportive comments.
- L 71- what is inflammatory proteins?
Response to Reviewer 1 - #1. Inflammatory proteins include c-reactive protein. We agree that it may be inconsistent and confusing to the reader as we refer to inflammatory biomarkers elsewhere in the manuscript. To be consistent with the language used in the manuscript, we have edited the statement to read:
“Chronic inflammation resulting in elevated levels of inflammatory biomarkers have been associated with a higher risk for the development of cardiovascular disease (CVD), diabetes, and other non-communicable diseases.”
- Introduction is well written, consistent.
- L 119 – the sources were properly chosen
- Study design selection is well described
- Authors have presented the definition of fructose-containing sugar
- Outcomes and data analysis were described separately
- there was even presented the certainty of the evidence
- L 192 – particular trials are described in details.
- Parts 2 and 3 are very informative in many details, supported by figures and table, all is enriched by supplementary material.
- The large supplementary material is prepared carefully. It took time.
- Discussion is written in reasonable manner. Taken into account the quantity of the material used, it is very consistent.
- in general, the authors deserve congratulations for such a good job and the time put into its preparation
Response to Reviewer 1 - #2-12. We are grateful for your review of our manuscript and positive comments.

Reviewer 2 Report
This study is to address the issue as to whether fructose is deleterious in humans. The reviewer agree that this is likely an important attempt given that we are surrounded by food high in fructose and therefore need answer as to whether or not those foods are safe. Author conducted a systemic review and meta-analysis of controlled feeding trials. As a result, they showed a predicted data that SSB could cause inflammation while natural fruits and dark chocolate could reduce it. While this study seems well done and provide important information, the reviewer think it might be better to put additional description particularly in the discussion.
A paper describes interesting points regarding the several issues of clinical study on fructose (Rheumatology 2019;58:1133-1141 https://doi.org/10.1093/rheumatology/kez128). Usually, this type of clinical study examining fruit consumption or food intake has difficulties to obtain data and quantitate the amount of fructose in human study. It is because of the data regarding to food intake could be obtained relying on the subject’s memory, lacking accuracy. Perhaps, the accuracy of data varies among individual study. Likewise, the amount of fructose would be also different among seasons, and countries. When it comes to the study with meta-analysis, it might be better to mention this issue as a limitation in this manuscript.
It is also interesting to discuss the role of uric acid since uric acid is likely a mediator for fructose induced metabolic syndrome. In this regard, a point would be whether serum uric acid is associated with SSB intake and fruit intake. If there is no data available, just a discussion would be appreciated.
Author Response
Please find below the Reviewer comments where we have responded to each comment as indicated in italicized font.
Reviewer #2:
This study is to address the issue as to whether fructose is deleterious in humans. The reviewer agree that this is likely an important attempt given that we are surrounded by food high in fructose and therefore need answer as to whether or not those foods are safe. Author conducted a systemic review and meta-analysis of controlled feeding trials. As a result, they showed a predicted data that SSB could cause inflammation while natural fruits and dark chocolate could reduce it. While this study seems well done and provide important information, the reviewer think it might be better to put additional description particularly in the discussion.
A paper describes interesting points regarding the several issues of clinical study on fructose (Rheumatology 2019;58:1133-1141 https://doi.org/10.1093/rheumatology/kez128). Usually, this type of clinical study examining fruit consumption or food intake has difficulties to obtain data and quantitate the amount of fructose in human study. It is because of the data regarding to food intake could be obtained relying on the subject’s memory, lacking accuracy. Perhaps, the accuracy of data varies among individual study. Likewise, the amount of fructose would be also different among seasons, and countries. When it comes to the study with meta-analysis, it might be better to mention this issue as a limitation in this manuscript.
Response to Reviewer 2 - #1. Thank you for sharing this publication and your thoughts on limitations to assessing dietary intake. The studies included in our meta-analysis were controlled trials, the majority of which provided the food source of fructose-containing sugar that was being investigated. We highlighted this point in our results when we discuss the percentage of trials in which the feeding control (the way in which the intervention was provided) was predominantly supplemented, which reads:
“Feeding control was mostly supplemented for substitution (77%), addition (96%), subtraction (100%), and ad libitum (100%) trials.”
Therefore, in the majority of trials in our meta-analysis, the amount of food source of fructose-containing sugar was less subject to recall bias regarding intake by the study participants as the foods were provided to them as prescribed.
It is also interesting to discuss the role of uric acid since uric acid is likely a mediator for fructose induced metabolic syndrome. In this regard, a point would be whether serum uric acid is associated with SSB intake and fruit intake. If there is no data available, just a discussion would be appreciated.
Response to Reviewer 2 - #2. We agree that the effect of fructose-containing sugars on uric acid is important as uric acid is a likely mediator for metabolic syndrome. As part of our series of systematic reviews and meta-analyses on the effects of different food sources of fructose-containing sugars on cardiometabolic outcomes, we previously published on the effects on uric acid (Ayoub-Charette S, et al. Different Food Sources of Fructose-Containing Sugars and Fasting Blood Uric Acid Levels: A Systematic Review and Meta-Analysis of Controlled Feeding Trials. J Nutr 2021;00:1–13.). Briefly, we found evidence of an interaction by food source in substitution and addition trials, where in substitution trials (energy matched) sugar sweetened beverages (SSBs) and sweets and desserts increased uric acid levels, while in addition trials (as a source of excess energy) SSBs increased and 100% fruit juice decreased uric acid levels. No significant effect was observed for fruit intake, however there were only 2 and 4 trials available for substitution and addition analyses, respectively. We agree with your suggestion and have added to our discussion including reference to the article you shared, which reads:
“Bioactives, including antioxidants and flavonoids, may also interfere with fructose metabolism. For example, antioxidants and flavonoids may reduce oxidative stress and thus fructose-induced uric acid production(144), which is supported by the lack of harm observed for fruit and fruit juice in a systematic review and meta-analysis of food sources of fructose-containing sugars on uric acid which contrasts the significant increases in uric acid that were observed for SSBs(8).”
